# POLICY PATH PROGRAMMING

## ABSTRACT

We develop a normative theory of hierarchical model-based policy optimization for Markov decision processes resulting in a full-depth, full-width policy iteration algorithm. This method performs policy updates which integrate reward information over all states at all horizons simultaneously thus sequentially maximizing the expected reward obtained per algorithmic iteration. Effectively, policy path programming ascends the expected cumulative reward gradient in the space of policies defined over all state-space paths. An exact formula is derived which finitely parametrizes these *path gradients* in terms of action preferences. Policy path gradients can be directly computed using an internal model thus obviating the need to sample paths in order to optimize in depth. They are quadratic in successor representation entries and afford natural generalizations to higher-order gradient techniques. In simulations, it is shown that intuitive hierarchical reasoning is emergent within the associated policy optimization dynamics.

## 1 INTRODUCTION

Reinforcement learning algorithms can leverage internal models of environment dynamics to facilitate the development of good control policies (Sutton & Barto, 2018). Dynamic programming methods iteratively implement one-step, full-width backups in order to propagate reward information across a state-space representation and then use this information to perform policy updates (Bellman, 1954). Stochastic approximations of this approach underpin a wide range of model-free reinforcement learning algorithms which can be enhanced by the ability to query samples from an "internal" environment model as in the DYNA architecture (Sutton, 1990). State-space search strategies apply heuristic principles to efficiently sample multi-step paths from internal models and have formed a core component of recent state-of-the-art game playing agents (Silver et al., 2016). Model-based policy search (Deisenroth & Rasmussen, 2011; Abdolmaleki et al., 2015) and gradient methods (Sutton et al., 1999) require sampled paths to approximate policy gradients based on either pure Monte Carlo estimation or by integrating long-run value estimates. All such methods rely on alternating between simulating paths over various horizons and then using this information to improve the policy either directly or indirectly by backing up value estimates and then inferring a policy (Sutton & Barto, 2018; Puterman, 1994). In this study, we introduce *policy path programming* (3P) which, given an environment model, normatively improves policies in a manner sensitive to the distribution of all future paths without requiring multi-step simulations. In particular, path programming follows the unique trajectory through policy space which iteratively maximizes the expected cumulative reward obtained. We develop 3P for entropy-regularized discounted Markov decision processes (Levine, 2018).

In the entropy-regularized MDP framework, a policy complexity penalty is added to the expected cumulative reward objective (Levine, 2018) (see Section 2 for details). Entropy regularization has several implications which have been investigated previously. The entropy penalty forces policies to be stochastic thereby naturally integrating an exploratory drive into the policy optimization process (Ahmed et al., 2018). In particular, the optimal policy can be immediately derived using calculus of variations as a Boltzmann-Gibbs distribution and reveals a path-based consistency law relating optimal value estimates and optimal policy probabilities which can be exploited to form a learning objective (Nachum et al., 2017). Furthermore, several studies have successfully used the relative entropy penalty to impose a conservative policy "trust region" to constrain policy updates thereby reducing erroneous policy steps due to the high variance in gradient estimation (Azar et al., 2012; Schulman et al., 2015). With this setup, we seek to compute this gradient exactly based on a consideration of

the distribution of all possible paths that the currently estimated policy will generate. Therefore, we express the MDP objective as a "sum-over-paths" and develop our model in this representation.

In the sum-over-paths formalism (Kappen, 2005; Theodorou et al., 2013), the central object of interest is not a state-action pair (Fig. 1A), as is the standard perspective in reinforcement learning in discrete MDPs, but complete state-action sequences or paths (Fig. 1B). The entropy-regularized cumulative reward objective can be re-written in terms of paths and a *path policy* can be expressed as an assignment of a probability to each path individually (see Section 3 for details). Gradient ascent in the space of policies over paths integrates information over all possible future paths in expectation at every step. 3P is defined as the gradient ascent algorithm which performs policy updates with respect to this *path gradient*. As a policy iteration method, we show that this is analogous to full-depth, full-width backups. Furthermore, we describe the associated natural path gradient which is fundamentally distinct from previous natural gradient techniques which utilize the asymptotic time limit of local, state-specific, natural policy gradients (Kakade, 2001; Peters et al., 2005). In Section 2, we summarize the mathematical framework of entropy-regularized MDPs from the path-based perspective and define our notation. In Section 3, we derive policy path programming. In Section 4, we apply the algorithm in state-spaces drawn from a variety of domains and analyze the resulting policy optimization dynamics. We conclude with a discussion in Section 5.

## 2    BACKGROUND AND NOTATION

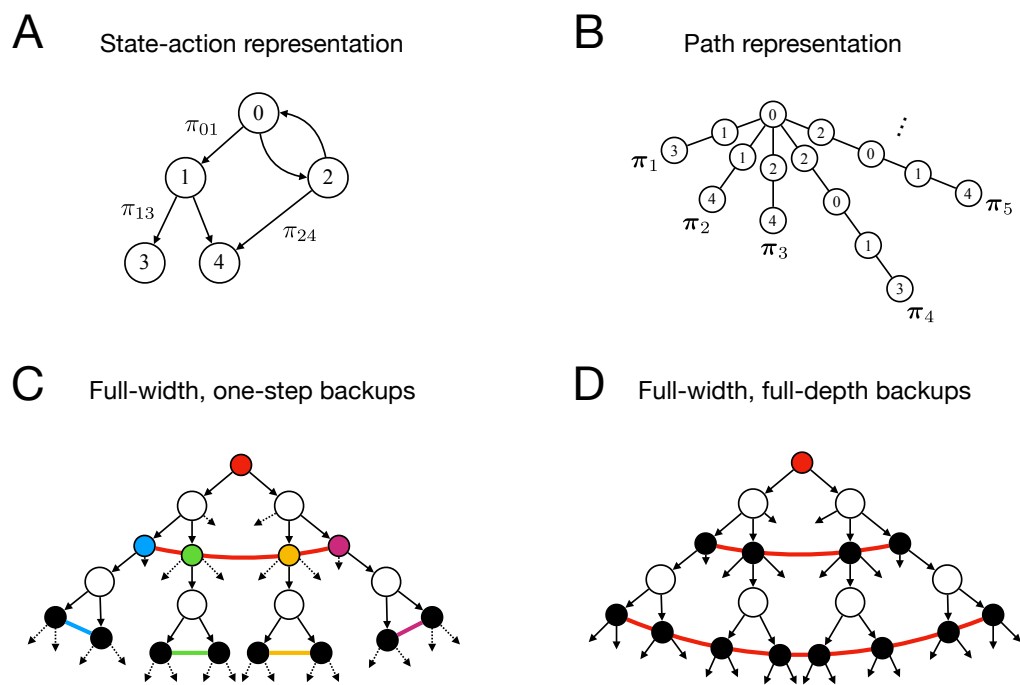

Figure 1: **A.** State-action representation of a simple state-space with recurrency and terminal states. **B.** A path space representation of the same state-space. **C.** A backup diagram (Sutton & Barto, 2018) where each full-width, one-step backup is color-coded by state. **D.** A backup diagram of the full-width, full-depth backups performed by policy path programming.

We develop the path programming formalism in the context of stochastic environmental dynamics. A state-space $\mathcal{X}$ is composed of states $x \in \mathcal{X}$ and the policy $\pi(a_j|s_i)$ describes the probability of selecting action $a_j$ in state $s_i$. The reward associated with transitioning from state $s_i$ to state $s_k$ after selecting action $a_j$ is denoted $\mathrm{R}(s_i, a_j, s_k) \equiv R_{ijk}$. Bold-typed notation, $\mathbf{s} \in \mathcal{S}$, $\mathbf{a} \in \mathcal{A}$, and $\mathbf{u} \in \mathcal{U}$ denotes sequences of states $s \in \mathcal{S}$, actions $a \in \mathcal{A}$, and state-action combinations $u \equiv (s_u, a_u) \in \mathcal{U}$ respectively. The action set $\mathcal{A}$ is the union of the sets of actions available at each state $\mathcal{A} = \cup_{s_i \in \mathcal{S}} \mathcal{A}_i$.

A valid state-action sequence $\mathbf{u} := (\ldots, \mathbf{s}_t, \mathbf{a}_t, \mathbf{s}_{t+1}, \mathbf{a}_{t+1}, \ldots)$ is referred to as a state-action *path*. The path probability $\mathbf{p}(\mathbf{u})$ is defined as the joint distribution over states $\mathbf{s}$ and actions $\mathbf{a}$

$$\mathbf{p}(\mathbf{u}) \quad := \quad \prod_{t=0}^{\infty} \mathrm{p}(\mathbf{s}_{t+1}|\mathbf{s}_t, \mathbf{a}_t)\pi(\mathbf{a}_t|\mathbf{s}_t) = \mathbf{p}(\mathbf{s}^{+1}|\mathbf{s}, \mathbf{a})\boldsymbol{\pi}(\mathbf{a}|\mathbf{s}) \tag{1}$$

where

$$\boldsymbol{\pi}(\mathbf{a}|\mathbf{s}) \quad := \quad \prod_{t=0}^{\infty} \pi(\mathbf{a}_t|\mathbf{s}_t) \quad , \quad \mathbf{p}(\mathbf{s}^{+1}|\mathbf{s}, \mathbf{a}) := \prod_{t=0}^{\infty} \mathrm{p}(\mathbf{s}_{t+1}|\mathbf{s}_t, \mathbf{a}_t) \ . \tag{2}$$

The environment dynamics are expressed in the function $\mathrm{p}(s_k|s_i, a_j)$ which denotes the probability of transitioning to state $s_k$ after selecting action $a_j$ in state $s_i$. The MDP objective as a sum-over-paths (Kappen, 2005; Theodorou et al., 2013) is

$$\boldsymbol{\pi}^* \quad := \quad \arg\max_{\boldsymbol{\pi}} \langle \mathbf{R}(\mathbf{u}) \rangle_{\mathbf{p}}$$

$$\mathbf{R}(\mathbf{u}) \quad = \quad \sum_{t=0}^{\infty} \mathrm{R}(\mathbf{s}_t, \mathbf{a}_t, \mathbf{s}_{t+1}) \tag{3}$$

where the angled brackets $\langle \cdot \rangle_{\mathbf{p}}$ denote the expectation operation over the path density $\mathbf{p}$. The form of the MDP objective in Equation 3 expresses a sequential decision-making problem as the determination a single decision but over paths. From this point of view, the max operation in Eqn. 3 is a full-width, full-depth policy iteration which converges in one step. We use the term full-depth because the paths incorporate information over all horizons (thus deep in time) and the term full-width because the max operation considers all paths (Fig. 1D). In contrast, policy iteration algorithms use full-width, one-step backups (Fig. 1C). This full-depth, full-width max operation is intractable since it requires a search over all possible paths and so we relax this problem using entropy regularization.

In the entropy-regularized reinforcement learning framework (Levine, 2018), a policy description length penalty for each path $\mathbf{u}$ weighted by a temperature parameter $\tau$ is added to the MDP objective (Eqn. 3). The relative entropy regularizer $D_{\mathrm{KL}}[\boldsymbol{\pi}||\boldsymbol{\pi}^0]$ measures policy complexity as the deviation from a prior (possibly non-uniform) policy $\boldsymbol{\pi}^0$ (Todorov, 2007; Kappen et al., 2012; Theodorou et al., 2013). In this case, the policy description length penalty is $-\tau \log \frac{\boldsymbol{\pi}(\mathbf{a}|\mathbf{s})}{\boldsymbol{\pi}^0(\mathbf{a}|\mathbf{s})}$ and the resulting entropy-regularized transition rewards $J_{ijk} := J(s_i, a_j, s_k)$, path rewards $\mathbf{J}(\mathbf{u})$, and policy objective $\mathcal{J}[\boldsymbol{\pi}]$ are then

$$J_{ijk} \quad := \quad \mathrm{R}_{ijk} - \tau \log \pi_{ij} + \tau \log \pi_{ij}^0$$

$$\mathbf{J}(\mathbf{u}) \quad := \quad \sum_{t=0}^{\infty} J(\mathbf{a}_t, \mathbf{s}_t, \mathbf{s}_{t+1})$$

$$= \quad \mathbf{R}(\mathbf{u}) - \tau \log \boldsymbol{\pi}(\mathbf{a}|\mathbf{s}) + \tau \log \boldsymbol{\pi}^0(\mathbf{a}|\mathbf{s})$$

$$= \quad \mathbf{R}(\mathbf{u}) - \tau \log \mathbf{p}(\mathbf{u}) + \tau \log \mathbf{p}^0(\mathbf{u})$$

$$\mathcal{J}[\boldsymbol{\pi}] \quad = \quad \langle \mathbf{J}(\mathbf{u}) \rangle_{\mathbf{p}}$$

$$= \quad -\tau D_{\mathrm{KL}}\left[\mathbf{p}(\mathbf{u})||\mathbf{p}^0(\mathbf{u})e^{\tau^{-1}\mathbf{R}(\mathbf{u})}\right] \ . \tag{4}$$

where we have made use of the compressed notation $\pi_{ij} \equiv \pi(a_j|s_i)$ and $\mathbf{p}^0(\mathbf{u}) := \mathbf{p}(\mathbf{s}^{+1}|\mathbf{s}, \mathbf{a})\boldsymbol{\pi}^0(\mathbf{a}|\mathbf{s})$.

From an information-theoretic point of view, the optimal policy $\boldsymbol{\pi}^*$ which maximizes $\mathcal{J}[\boldsymbol{\pi}]$ gives the best trade-off between maximizing reward and minimizing policy encoding costs. An implication of the description length penalty is that encoding deterministic transitions is infinitely costly $[\log \boldsymbol{\pi}(\mathbf{a}|\mathbf{s}) \to -\infty$ as $\boldsymbol{\pi}(\mathbf{a}|\mathbf{s}) \to 0]$ and therefore the optimal policy will be stochastic. Taking the temperature parameter to zero $\tau \to 0$ recovers the standard MDP problem of identifying a deterministic policy in pursuit of maximum expected cumulative reward.

In the main text, policy path programming (3P) is developed for entropy-regularized MDPs with stochastic environment dynamics. It is straightforward to derive analogous update equations in the presence of deterministic environmental transitions which correspond to the subset of control problems known as KL-control (Kappen et al., 2012) or linearly-solvable Markov decision processes (Todorov, 2007)). Furthermore, our analysis can be applied to MDPs with absorbing states. Thus, path programming can be applied to a broad class of MDPs.

## 3    POLICY PATH PROGRAMMING IN DISCRETE MARKOV DECISION PROCESSES

The policy objective function (Eqn. 4) can be re-expressed as

$$
\mathcal{J}[\boldsymbol{\pi}] \;=\; \sum_{\mathbf{u}\in\mathcal{U}} \prod_{\substack{s_i,s_k\in\mathcal{S}\\ a_j\in\mathcal{A}_i}} (\pi_{ij}p_{ijk})^{n_{ijk}(\mathbf{u})} \left\{ \sum_{\substack{s_i,s_k\in\mathcal{S}\\ a_j\in\mathcal{A}_i}} n_{ijk}(\mathbf{u})\left[R_{ijk}-\tau\left(\log\pi_{ij}-\log\pi_{ij}^0\right)\right] \right\}
$$

$$
\text{s.t.}\quad \pi_{ij}>0 \quad \forall s_i\in\mathcal{S}, a_j\in\mathcal{A}_i \;,\; \sum_{a_j\in\mathcal{A}}\pi_{ij}=1 \quad \forall s_i\in\mathcal{S} \tag{5}
$$

where we have expressed the objective (Eqn. 4) in terms of *counters* $n_{ijk}(\mathbf{u})$ which quantify the number of times that $s_k$ is occupied after selecting action $a_j$ in state $s_i$ on path $\mathbf{u}$. We reparametrize the policy parameters $\pi_{ij}$ in terms of natural parameters $A_{ij}$ in an exponential model $\pi_{ij}:=e^{A_{ij}}$ (Nagaoka, 2005). These natural parameters are examples of action preferences in reinforcement learning parlance[1] (Sutton & Barto, 2018) and can take any negative real value. Substituting $A_{ij}:=\log\pi_{ij}$,

$$
\mathcal{J}[\mathbf{A}] \;=\; \sum_{\mathbf{u}\in\mathcal{S}} e^{\mathbf{A}\cdot\mathbf{n}(\mathbf{u})+\mathbf{C}\cdot\mathbf{n}(\mathbf{u})} \left[ \sum_{\substack{s_i,s_k\in\mathcal{X}\\ a_j\in\mathcal{A}_i}} n_{ijk}(\mathbf{u})\left(R_{ijk}-\tau A_{ij}+\tau A_{ij}^0\right) \right]
$$

$$
e^{\mathbf{A}\cdot\mathbf{n}(\mathbf{u})} \;=\; e^{\sum_{s_i,a_j,s_k} A_{ij}n_{ijk}(\mathbf{u})} = e^{\sum_{s_i,a_j} A_{ij}n_{ij}(\mathbf{u})}
$$

$$
e^{\mathbf{C}\cdot\mathbf{n}(\mathbf{u})} \;=\; e^{\sum_{s_i,a_j,s_k} C_{ijk}n_{ijk}(\mathbf{u})} \tag{6}
$$

where $C_{ijk}:=\log p_{ijk}$ and $A_{ij}^0:=\log\pi_{ij}^0$ has been analogously substituted, and $\mathbf{n}$ is a tensor with components $n_{ijk}$ and $[\mathbf{A}]_{ij}:=A_{ij}$ have been used for the event counters and action preferences respectively. Considering the set of probabilities $e^{(\mathbf{A}+\mathbf{C})\cdot\mathbf{n}(\mathbf{u})}$ parametrized by $\mathbf{A}$ as an exponential family (Nagaoka, 2005), the vector $\mathbf{n}$ of transition counters $n_{ijk}(\mathbf{u})$ constitutes a sufficient statistic for the path $\mathbf{u}$. Given that the policy transition probabilities $\pi_{ij}=e^{A_{ij}}$ are drawn from the action preferences $A_{ij}\in\mathbb{R}^-$ via an exponential transformation, we are guaranteed that $0<\pi_{ij}\le 1$ for all state-action combinations.

In order to ensure that $\pi^t$ always forms a probability distribution at every state, we eliminate a redundant action preference at each state. This is accomplished by defining an arbitrary transition probability at each state in terms of the probabilities of alternative transitions at that state. We index this dependent action preference using an $\omega$ subscript as in $A_{ii_\omega}$ in order to distinguish it from the independent action preferences which will be directly modified during policy optimization. We define $i_\omega$ as the action index of an arbitrary action available in state $s_i$. Under the local policy normalization constraint, the action preferences are equivalently constrained via

$$
A_{ii_\omega} \;=\; \log\left(1-\sum_{a_{i_\omega}\neq a_j\in\mathcal{A}_i} e^{A_{ij}}\right) \;. \tag{7}
$$

### 3.1    PATH GRADIENT CALCULATION

The goal is to iteratively update the action preferences $\mathbf{A}^t$ characterizing the current policy by gradient descent

$$
\mathbf{A}^{t+1} \;\leftarrow\; \mathbf{A}^t + \alpha\mathcal{I}^{-1}\nabla_{\mathbf{A}}\mathcal{J}\left[\mathbf{A}^t\right] \tag{8}
$$

where $\mathcal{I}$ is the Fisher information of the path probability density which naturalizes the gradient, and $\alpha$ is the stepsize. The partial derivatives underpinning the path policy gradient are derived using Corollary B.3.1 and Corollary B.2.1 which can be found in Section B of the Supplementary Material (SM).

---

[1]In particular, these action preferences converge to optimal advantage values (Levine, 2018).

**Theorem 3.1.** The policy path gradient in the exponential parametrization is defined by the partial derivatives

$$\partial_{A_{ij}} \mathcal{J}[\mathbf{A}] \;=\; \sum_{\substack{s_k \in \mathcal{S} \\ a_l \in \mathcal{A}_k}} \left[ \mathcal{C}_{ij,kl} - e^{A_{ij} - A_{ii_\omega}} \mathcal{C}_{ii_\omega, kl} \right] J_{kl} \tag{9}$$

where $\mathcal{C}_{ij,kl} := \langle n_{ij}(\mathbf{u}) n_{kl}(\mathbf{u}) \rangle_{\mathbf{p}}$ are state-action correlation functions and $J_{kl} := \langle J(s_i, a_j, s_k) \rangle_{\mathrm{P}(s_k | s_i, a_j)}$.

*Proof.*

$$\partial_{A_{ij}} \mathcal{J}[\mathbf{A}] = \partial_{A_{ij}} \left[ \sum_{\mathbf{u} \in \mathcal{U}} \mathbf{p}(\mathbf{u}) \mathbf{J}(\mathbf{u}) \right]$$

$$= \sum_{\mathbf{u} \in \mathcal{U}} \left[ \partial_{A_{ij}} \mathbf{p}(\mathbf{u}) \right] \mathbf{J}(\mathbf{u}) + \sum_{\mathbf{u} \in \mathcal{U}} \mathbf{p}(\mathbf{u}) \left[ \partial_{A_{ij}} \mathbf{J}(\mathbf{u}) \right]$$

$$= \sum_{\mathbf{u} \in \mathcal{U}} \left[ \partial_{A_{ij}} \mathbf{p}(\mathbf{u}) \right] \mathbf{J}(\mathbf{u}) \qquad\qquad (\Leftarrow \text{Corollary B.3.1})$$

$$= \sum_{\mathbf{u} \in \mathcal{U}} \left[ \mathbf{p}(\mathbf{u}) \left[ n_{ij}(\mathbf{u}) - e^{A_{ij} - A_{ii_\omega}} n_{ii_\omega}(\mathbf{u}) \right] \right] \mathbf{J}(\mathbf{u}) \qquad (\Leftarrow \text{Corollary B.2.1})$$

$$= \sum_{\mathbf{u} \in \mathcal{U}} \left\{ \mathbf{p}(\mathbf{u}) \left[ n_{ij}(\mathbf{u}) - e^{A_{ij} - A_{ii_\omega}} n_{ii_\omega}(\mathbf{u}) \right] \right\} \left[ \sum_{\substack{s_k, s_m \in \mathcal{S} \\ a_l \in \mathcal{A}_k}} n_{klm}(\mathbf{u}) J_{klm} \right]$$

$$= \sum_{\substack{s_k, s_m \in \mathcal{S} \\ a_l \in \mathcal{A}_k}} \left[ \langle n_{ij}(\mathbf{u}) n_{klm}(\mathbf{u}) \rangle_{\mathbf{p}} - e^{A_{ij} - A_{ii_\omega}} \langle n_{ii_\omega}(\mathbf{u}) n_{klm}(\mathbf{u}) \rangle_{\mathbf{p}} \right] J_{klm}$$

$$= \sum_{\substack{s_k, s_m \in \mathcal{S} \\ a_l \in \mathcal{A}_k}} \left[ \mathcal{C}_{ij,kl} - e^{A_{ij} - A_{ii_\omega}} \mathcal{C}_{ii_\omega, kl} \right] p_{klm} J_{klm}$$

$$= \sum_{\substack{s_k \in \mathcal{S} \\ a_l \in \mathcal{A}_k}} \left[ \mathcal{C}_{ij,kl} - e^{A_{ij} - A_{ii_\omega}} \mathcal{C}_{ii_\omega, kl} \right] J_{kl} \;\; .$$

□

A closed-form expression for the state-action correlations $\mathcal{C}_{ij,kl}$ is derived using Markov chain theory (Kemeny & Snell, 1983). The Fisher information $\mathcal{I}$ with respect to the path density is calculated in Section B.2 (SM).

## 3.2 Algorithm summary and intuition

Based on these derivations, the policy path programming algorithm in the exponential parametrization which implements the updates in Eqn. 8 is:

$$
\begin{aligned}
\pi_{ij}^t &:= e^{A_{ij}^t} \\
T_{ij}^t &:= \sum_{a_{i'} \in \mathcal{A}_i} \pi_{ii'j}^t p_{ii'j} \quad \forall s_i, s_j \in \mathcal{X} \\
D_{ij}^t &= \left[ (I - \lambda T)^{-1} \right]_{ij} \\
E_{(ij)k}^t &:= \left[ P D^t \right]_{(ij)k} \\
\mathcal{C}_{ij,kl}^t &= D_{\emptyset i}^t e^{A_{ij}^t} \delta_{ik} \delta_{jl} + \left[ D_{\emptyset i}^t E_{(ij)k}^t + D_{\emptyset k}^t E_{(kl)i}^t \right] e^{A_{ij}^t} e^{A_{kl}^t} \\
\mathcal{I}_{ij,kl}^t &= \mathcal{C}_{ij,kl}^t - e^{A_{kl}^t - A_{kk_\omega}^t} \mathcal{C}_{ij,kk_\omega}^t - e^{A_{ij}^t - A_{ii_\omega}^t} \mathcal{C}_{kl,ii_\omega}^t + e^{A_{ij}^t + A_{kl}^t - A_{ii_\omega}^t - A_{kk_\omega}^t} \mathcal{C}_{ii_\omega,kk_\omega}^t \\
J_{kl}^t &= R_{kl} - \tau A_{kl}^t + \tau A_{kl}^0 \\
A_{ij}^{t+1} &\leftarrow A_{ij}^t + \alpha \sum_{x_m, x_n \in \mathcal{X}} \left[ \mathcal{I}_{mn,ij}^t \right]^{-1} \left\{ \sum_{s_k, x_l \in \mathcal{X}} \left[ \mathcal{C}_{ij,kl}^t - e^{A_{ij}^t - A_{ii_\omega}^t} \mathcal{C}_{ii_\omega,kl}^t \right] J_{kl}^t \right\} \qquad (10) \\
A_{ii_\omega}^{t+1} &= \log \left( 1 - \sum_{x_{i_\omega} \neq s_j \in \mathcal{X}_i} e^{A_{ij}^{t+1}} \right) .
\end{aligned}
$$

where $\lambda$ is a free parameter controlling the agent's "foresight" or how far into the future it can "see". 3P requires that $0 \lambda < 1$ in order to ensure that the components of the path gradient expression converge to finite quantities. This parameter can also be conceptualized as a standard reward discount parameter $\gamma$ as in discounted MDPs. Note that the regularized transition reward $J$, transient transition matrix $T$, successor representation $D$, Fisher information $\mathcal{I}$, and counter correlations $\mathcal{C}$, all depend on the current policy estimate $\pi^t$. The initialization of action preferences $A_{ij}^0$ is discussed in the SM (subsection B.3). In all simulations, we fix the foresight $\lambda = 0.99$ (thus simulating an "expert" planner with "deep" foresight), the stepsize $\alpha = 0.001$ (chosen such that 3P tracked the policy evolution at high precision for the purposes of visualization), and the temperature $\tau = 1$ (taking the natural default parameter). In future work, we will explore the implications of reducing $\lambda$ to simulate a planner with short "foresight" and using the path Hessian to optimize $\alpha$. The temperature $\tau$ controls the policy stochasticity which has been explored previously in model-free (Ahmed et al., 2018) and model-based (Azar et al., 2012) reinforcement learning.

The path gradient (Eqn. 10) has several intuitive properties. For each state, it backups rewards from all other states based on all future paths thus implementing a full-depth, full-width update from a dynamic programming point of view (Sutton & Barto, 2018). The matrix $D$ is the successor representation (Dayan, 1993). An entry $D_{ij}$ counts the expected number of times that state $s_j$ will be occupied after starting from state $s_i$. Therefore the counter correlations $\mathcal{C}$, which is quadratic in successor representations, reflect the rate of co-occurrence of pairs of state-actions on average under the policy-generated path distribution. This enables the algorithm to understand the correlative structure of state occupations under the current policy. For example, if a temporally remote action $s_k \to x_l$ has high reward $J_{kl}$ and if there is a high counter correlation $\mathcal{C}_{ij,kl}$ between a local action $s_i \to s_j$ and the remote action (over all horizons), then the reward $J_{kl}$ associated with the remote action will be weighted heavily in the path gradient and added to the local action preference $A_{ij}$. The magnitude of this backup is explicitly normalized with respect to a baseline counter correlation $\mathcal{C}_{ii_\omega,kl}$ associated with the dependent action preference. That is, if the action $s_i \to x_{i_\omega}$ is also strongly correlated with $s_k \to x_l$ then the backup to $A_{ij}$ is attenuated since the unique contribution of $s_i \to s_j$ in generating $s_k \to x_l$ is diminished. Using such attributional logic, path programming updates action preferences based on the degree to which a state-action independently leads to rewarding state-space paths over all depths.

## 4 SIMULATIONS

We simulate path programming (Eqn. 10) in a variety of simple reinforcement learning environments in order to gain insight into the dynamics of the policy iteration process.

### 4.1 ANALYSIS

After running policy path programming until convergence, its dynamics are interrogated using two measures. The first measure is the KL-divergence between the policy densities at each iteration $\pi^t$ and the prior policy $\pi^0$. We compute this *policy divergence* measure PD locally at each state $x \in \mathcal{X}$:

$$\mathrm{PD}(x,t) := D_{\mathrm{KL}} \left[ \pi_{x\cdot}^t || \pi_{x\cdot}^0 \right] \tag{11}$$

Policy divergence quantifies the degree to which the algorithm is modifying the local policy at each state as a function of planning time. The second measure is the difference between the expected number of times a state will be occupied under the currently optimized policy versus the prior policy. Specifically, the *counter difference* measure CD is

$$\mathrm{CD}(x,t) := D_{\emptyset x}^t - D_{\emptyset x}^0 \ . \tag{12}$$

where $x_\emptyset$ is the initial state. Counter differences shows how path programming prioritizes the occupation of states in time. We study these measures as well as their time derivatives in their original form as well as after max-normalizing per state in order to facilitate comparisons across states:

$$\widetilde{\mathrm{PD}}(x,t) := \frac{\mathrm{PD}(x,t)}{\max_t \mathrm{PD}(x,t)} \ , \ \widetilde{\mathrm{CD}}(x,t) := \frac{\mathrm{CD}(x,t)}{\max_t |\mathrm{CD}(x,t)|} \ . \tag{13}$$

### 4.2 EXPERIMENTS

We implement path programming in decision trees (Fig. 2 and Fig. S1, SM), the Tower of Hanoi problem (Fig. 3 and Fig. S2, SM), and four-room grid worlds with and without a wormhole (Fig. 4 and Fig. S4, SM). The decision tree example shows how path programming optimizes with respect to the path structure of the environment, the Tower of Hanoi example highlights its intuitive hierarchical qualities, and, in the grid worlds, the capacity of 3P to radically alter its dynamics in response to the state-space modifications is observed.

In the decision tree environments, 3P implements a backward induction strategy from the terminal goal node to the initial state along the optimal path. Path programming increases the probability of the agent moving along the optimal path only and leaves all other paths untouched throughout the policy optimization process. The added decision complexity at state 2 Fig. 2A is reflected in the total policy divergence at that state and consequently the time-to-peak as compared to the other states along the optimal path.

In our Tower of Hanoi simulation (Fig. 3), the agent is endowed with the ability to remain at a state thus the optimal policy is to transit to state G and then choose to remain there (since it can then accumulate a reward on every time step). When path programming, the agent prioritizes the adaptation of its policy so that it remains at the goal state. This can be observed in the relatively rapid policy divergence[2] $\widetilde{\mathrm{PD}}$ at the goal state (Fig. 3B) and the fact that the policy divergence velocity peaks for the goal state before all others (Fig. 3D). The second highest priority is assigned to bottleneck states along the optimal path. The optimization of the local policy at the start state is deferred to last. Through the counter difference measure $\widetilde{\mathrm{CD}}$, we can observe how path programming increases the occupation density of all states in the same cluster as the goal state (in blue) before subsequently reducing the occupation density of non-goal states in the goal cluster (Fig. 3E). These non-monotonic counter difference trajectories suggest that path programming treats all blue states as a single unit initially before refining its planning strategy to distinguish individual states within the goal cluster. Increasing the resolution at which it distinguishes states over time as well as prioritizing local policy adaptations starting with the goal state through the bottleneck states and ending with the start state, suggests that path programming is sensitive to the hierarchical structure of the state-space. In the SM,

---

[2]Here, we present normalized policy divergence curves to facilitate comparisons across states. The equivalent unnormalized curves may be found in the Fig. S2, SM.

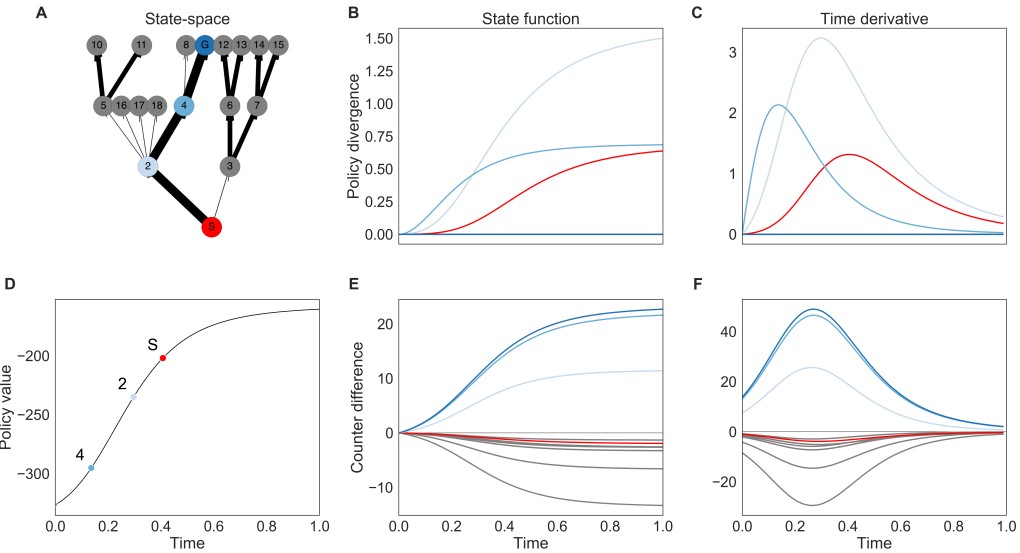

Figure 2: **Decision tree with added decision complexity.** Panels as in Fig. 3. A higher local policy divergence at state 2 is observed (as compared to Fig. S1).

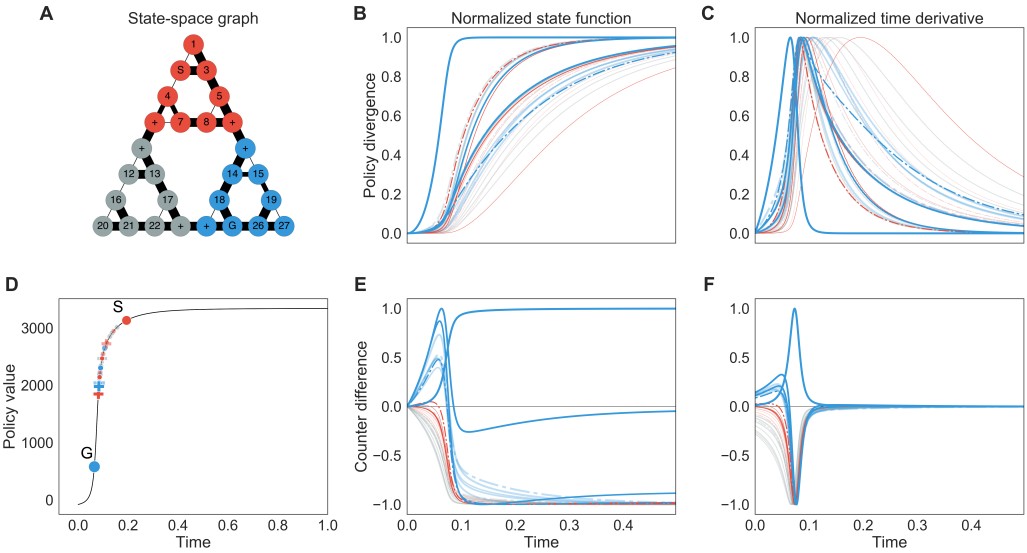

Figure 3: **Path programming the optimal policy in the Tower of Hanoi game. A.** Tower of Hanoi state-space graph. **B-C.** Normalized policy divergence $\widetilde{\text{PD}}$ and its time derivative for each state. The color of the curve indicates which state it corresponds to in panel A. Dotted lines correspond to bottleneck states marked + in panel A. Lines for states which are not along the optimal path are plotted transparently. **D.** Policy value as a function of planning time. Time-to-max policy divergence velocities (i.e. the peaks of the curves in panel C) are dotted along the policy value curve for states along the optimal path. **E-F.** Normalized counter difference $\widetilde{\text{CD}}$ and its time derivative.

we present the results of path programming under an alternative scenario whereby the agent is reset back to the start state on arrival at the goal (Fig. S3, SM).

In the room world simulation (Fig. 4), the agent must navigate from the start state S in the northwest room to the goal state G in the southeast room (panel A). It can do so via a path through the other

rooms or, for the shortest route, step through the "wormhole" W from the northwest room directly to the southeast room. We compare policy path programming in this scenario against the same scenario but with the wormhole removed (Fig. S4, SM). Despite the relatively minor modification to the transition structure of the state-space, policy path programming restructures its processing with the key distinction being that policy path programming prioritizes the wormhole at the earliest stages of processing. Specifically, the policy at the wormhole entrance initially diverges most rapidly from its prior policy (Fig. 4B, red line, long dashes) is due to the steepest acceleration in PD (Fig. 4C). Conversely, the wormhole exit is prioritized based on the counter difference measure CD (Fig. 4E, blue line, long dashes). This shows that path programming begins with policy improvements which ensure that the agent makes use of the wormhole.

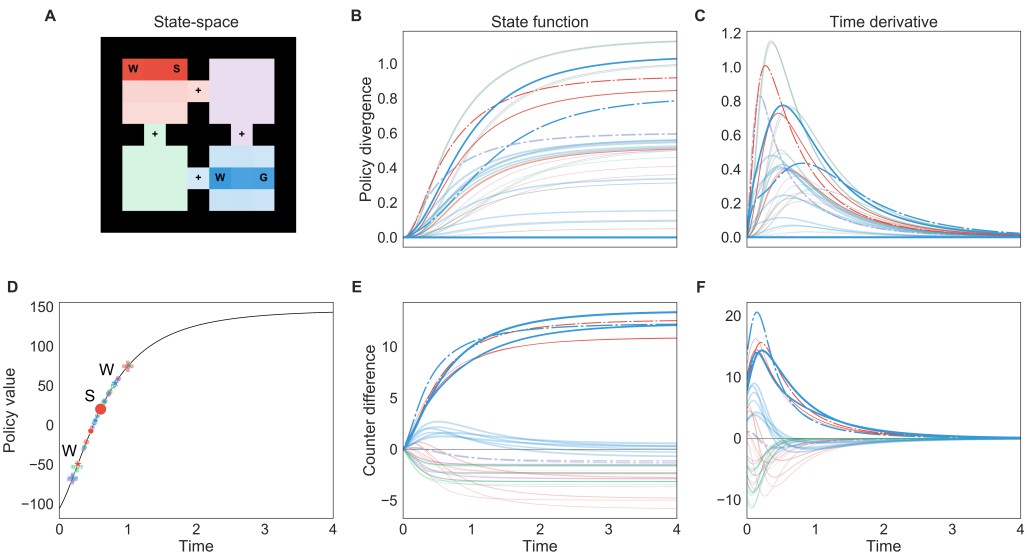

Figure 4: **Path programming the optimal policy in a grid world with a wormhole.** Panels as in Fig. 3. Dotted lines with short dashes correspond to bottleneck states marked + in panel **A**. Dotted lines with long dashes correspond to wormhole states marked W in panel A. The darkness of the state coloring reflects state occupation density under the optimal policy.

## 5 DISCUSSION

We introduced a novel natural gradient procedure sensitive to the on-policy path density. If the environmental model is known, then this gradient can be computed in analytically. As a policy iteration procedure, policy path programming implements full-depth, full-width backups in contrast to other dynamic programming methods (operating on tabular representations) which use one-step, full-width backups (Sutton & Barto, 2018). In previous work, natural policy gradient and actor-critic methods (Kakade, 2001; Bagnell & Schneider, 2003; Peters et al., 2005) have modified standard policy search methods using Fisher information matrices in order to perform policy updates in a manner that is sensitive to the KL-divergence between old and new local policies on average at each state. However, the definition of the natural path gradient used in these studies diverges from that elucidated here in a crucial way. They define the Fisher information matrix asymptotically in time which converges to the average of the local natural policy gradients at each state weighted by the induced stationary state distribution. This implies that these Fisher information matrices do not relate the parametrization of the policy gradient across time as in our method and thus is agnostic to the structure of the state-space. Indeed, in the action preference parametrization used here, the time-asymptotic Fisher information matrix will be diagonal. Though this time-asymptotic method is the only way to define a convergent metric for infinite horizon MDPs, it is not necessary for discounted (or episodic) MDPs as revealed in this study. The specific natural path gradient introduced here results in a hierarchical model-based policy optimization which, we suggest, may serve as a normative process model of optimal planning.

Policy path programming may be leveraged as a theoretic tool for analyzing the hierarchical structure of policy space since functional relationships between actions over all spatiotemporal scales are explicitly embedded within policy path gradients. This can be observed in the policy optimization dynamics generated by policy path programming. In the classic hierarchical tasks simulated here, path programming implicitly prioritizes policy improvements at critical bottleneck states, the evolution of occupation densities over states are dynamically clustered then distinguished (Fig. 3), and the policy evolution can be restructured in order to take advantage of shortcuts when available at the earliest stages of processing (Fig. 4). Whereas these effects manifest the output of path programming, it may be informative to explore the internal dynamical structure of path programming by analyzing how the counter correlation functions evolve over time.

As with other dynamic programming methods, path programming does not scale however it may provide some insights for developing novel scalable algorithms. For example, the path gradient components $\left[ \mathcal{C}_{ij,kl} - e^{A_{ij} - A_{ii_\omega}} \mathcal{C}_{ii_\omega,kl} \right] J^t_{kl}$ (Eqn. 10) may form an alternative, potentially more stable, objective for reinforcement learning based on path consistency (Nachum et al., 2017) since they will equal zero only at the globally optimal policy. Furthermore, the state-action counter correlation functions $\mathcal{C}_{ij,kl}$ may integrated with function approximation methods in order to derive value representations which linearize the natural path gradient in a manner analogous to asymptotic natural gradient methods Kakade (2001). Indeed, algorithms already designed to learn function approximators based on the successor representation could be adapted to this purpose (Barreto et al., 2016). While the successor representation facilitates the rapid evaluation of a policy, path gradients enable one to immediately improve a policy. In this respect, path programming reflects a shift from a representation learning strategy based on policy evaluation (Dayan, 1993) to one based on policy improvement. Importantly, policy path gradients exhibit the key successor representation property of decoupling the environment representation from the reward function and thus the same correlation functions can be flexibly transferred across tasks.

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

# Appendix

## CONTENTS

## A    EXTENDED SIMULATIONS AND ANALYSIS

### A.1    DECISION TREES

In Fig. S1 (main text), and Fig. 2, we apply path programming to a series of decision trees of increasing complexity. The agent acquires a reward of 10 points on arrival at the goal state G and is then teleported back to the start state S in order to play again.

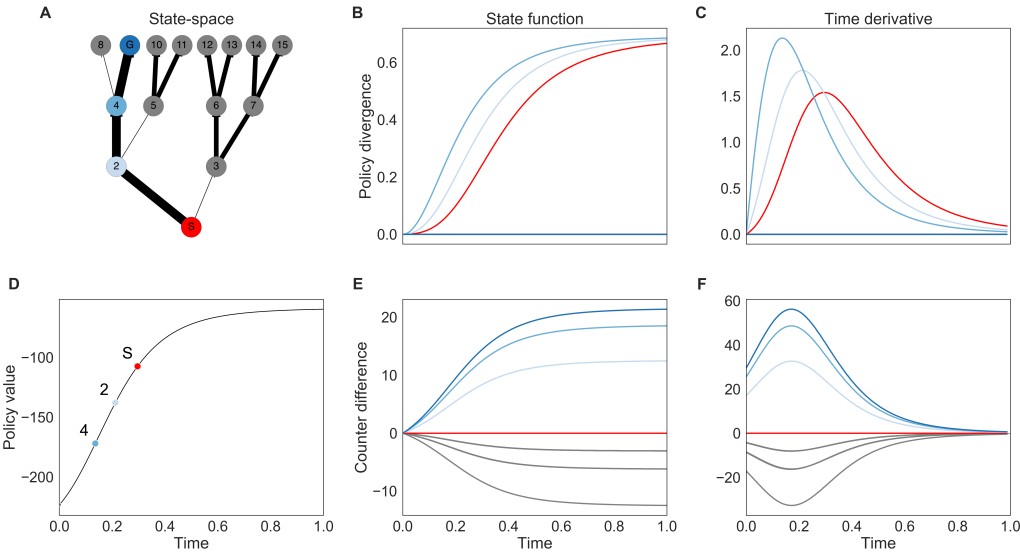

Figure S1: **Breadth two, depth three decision tree.** Panels as in Fig. 3. **A.** The state-space graph of a breadth two, depth three decision tree. States on the optimal path are highlighted in blue. Edge thickness reflects the optimal policy. According to the normalized policy divergences $\widetilde{\mathrm{PD}}$, path programming prioritizes policy optimization in state 4, then state 2, and then the start state S. This is reminiscent of a backward induction strategy. The counter differences $\widetilde{\mathrm{CD}}$ shows that path programming smoothly increases the probability that the agent will occupy the optimal path at the expense of all other paths.

## A.2 TOWER OF HANOI

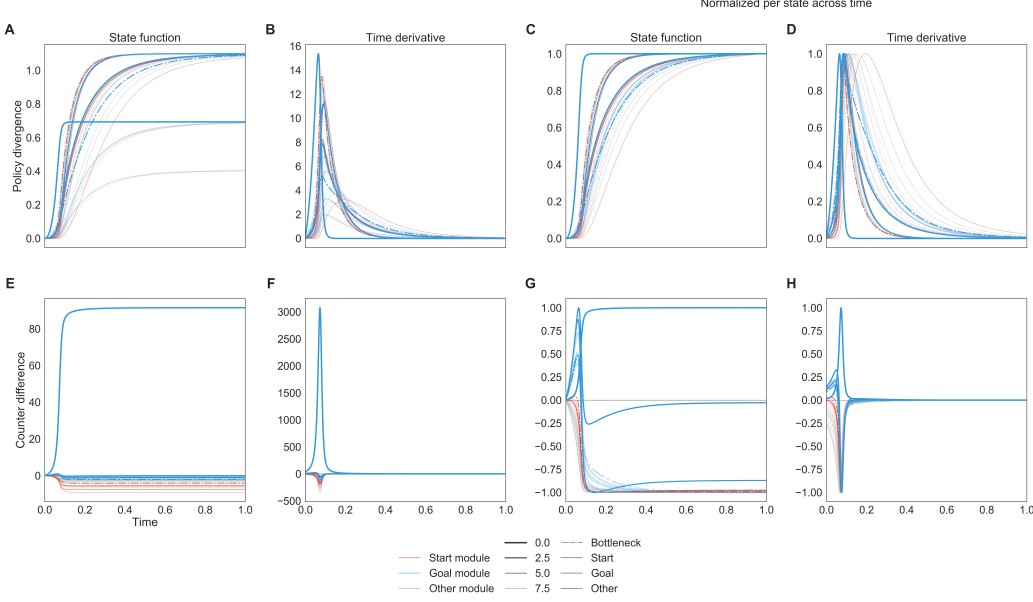

Figure S2: **Tower of Hanoi with the option to remain at a state.** Panels as in Fig. 3. We present an extended set of results. Panels C, D, G, and H have already been displayed in Fig. 3 while panels A, B, E, and F show their unnormalized counterparts.

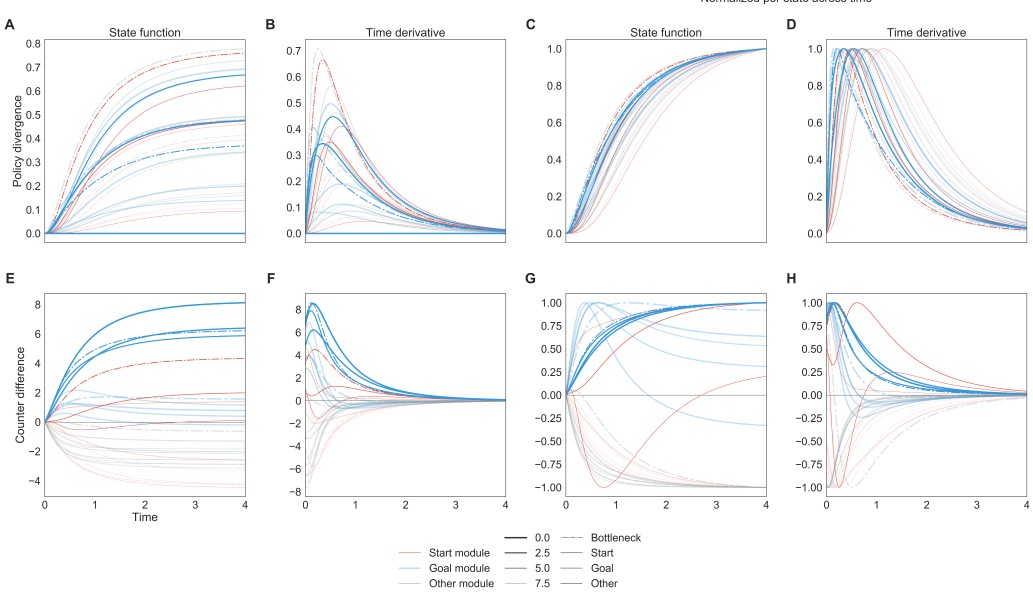

Figure S3: **Tower of Hanoi with forced resets on arrival at the goal.** Instead of having the option to remain at a goal state, we consider an alternative scenario in which the agent is automatically transported back to the initial state on after arriving at the goal. The path gradient dynamics are broadly similar and retain their hierarchical characteristics however the prominence of the goal state is diminished both in terms of policy divergence (since the agent no longer has any choice at the goal state) and counter difference (since they goal state can no longer be repeatedly exploited for reward).

## A.3  ROOM WORLD

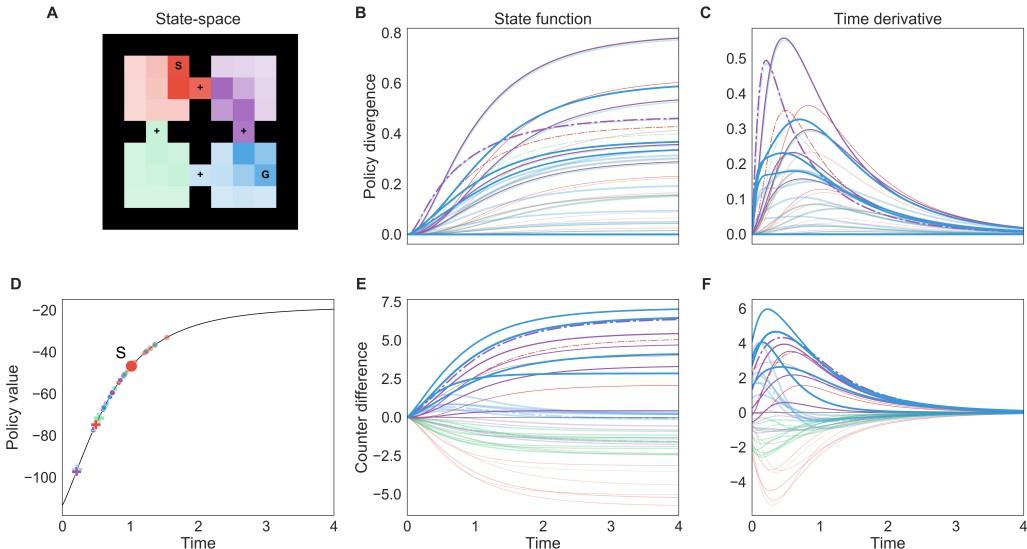

Figure S4: **Path programming the optimal policy in a grid world without a wormhole.** Panels as in Fig. 3. Dotted lines with short dashes correspond to bottleneck states marked + in panel **A**. Dotted lines with long dashes correspond to wormhole states marked W in panel **A**. Darker state colors indicate higher densities of state occupation under the optimal policy.

## B   POLICY PATH PROGRAMMING IN THE EXPONENTIAL PARAMETRIZATION

We derive results which are used to compute the gradient of $\mathcal{J}[\pi]$ (Eqn. 6) with respect to the parameters $A_{ij} = \log \pi_{ij}$ in the main text.

### B.1   PRELIMINARY CALCULATIONS

In this section, we record several complementary calculations.

**Proposition B.1.** The partial derivative $\partial_{A_{ij}} A_{kl}$ of an independent action preference $A_{kl}$ with respect to another independent action preference $A_{ij}$ is

$$\partial_{A_{ij}} A_{kl} = \delta_{ik}\delta_{kl} := \delta_{ij,kl} \ . \tag{14}$$

The partial derivative $\partial_{A_{ij}} A_{kk_\omega}$ of a dependent action preference $A_{kk_\omega}$ with respect to an independent action preference $A_{ij}$ is

$$\partial_{A_{ij}} A_{kk_\omega} = -\delta_{ik} e^{A_{kj} - A_{kk_\omega}} \ . \tag{15}$$

*Proof.* Eqn. 14 follows by definition. For Eqn. 15, we recall the constraint equation for dependent action preferences

$$
\begin{aligned}
\partial_{A_{ij}} A_{kk_\omega} &= \partial_{A_{ij}} \left[ \log \left( 1 - \sum_{a_{k_\omega} \neq a_l \in \mathcal{A}_k} e^{A_{kl}} \right) \right] \\
&= \left( 1 - \sum_{a_{k_\omega} \neq a_l \in \mathcal{A}_k} e^{A_{kl}} \right)^{-1} \delta_{ik} \left[ -e^{A_{kj}} \right] \\
&= -\delta_{ik} e^{A_{kj} - A_{kk_\omega}} \ .
\end{aligned}
\tag{16}
$$

$\square$

**Proposition B.2.** The partial derivatives of the log path density $\log \mathbf{p}(\mathbf{u})$ and log path policy $\log \boldsymbol{\pi}(\mathbf{u})$ with respect to action preference $A_{ij}$ is

$$
\begin{aligned}
\partial_{A_{ij}} [\log \mathbf{p}(\mathbf{u})] &= \partial_{A_{ij}} [\log \boldsymbol{\pi}(\mathbf{u})] \tag{17} \\
&= n_{ij}(\mathbf{u}) - e^{A_{ij} - A_{ii_\omega}} n_{ii_\omega}(\mathbf{u}) \ . \tag{18}
\end{aligned}
$$

*Proof.*

$$
\begin{aligned}
\partial_{A_{ij}} \log \mathbf{p}(\mathbf{u}) &= \partial_{A_{ij}} \left[ \log \boldsymbol{\pi}(\mathbf{a}|\mathbf{s}) + \log \mathbf{p}(\mathbf{s}^{+1}|\mathbf{s}, \mathbf{a}) \right] \\
&= \partial_{A_{ij}} \left[ \mathbf{A} \cdot \mathbf{n}(\mathbf{u}) \right] \\
&= \partial_{A_{ij}} \left\{ \sum_{s_k \in \mathcal{S}} \left[ \sum_{a_{k_\omega} \neq a_l \in \mathcal{A}_k} A_{kl} n_{kl}(\mathbf{u}) + A_{kk_\omega} n_{kk_\omega}(\mathbf{u}) \right] \right\} \\
&= \sum_{s_k \in \mathcal{S}} \left[ \sum_{a_{k_\omega} \neq a_l \in \mathcal{A}_k} \left( \partial_{A_{ij}} A_{kl} \right) n_{kl}(\mathbf{u}) + \left( \partial_{A_{ij}} A_{kk_\omega} \right) n_{kk_\omega}(\mathbf{u}) \right] \\
&= n_{ij}(\mathbf{u}) - e^{A_{ij} - A_{ii_\omega}} n_{ii_\omega}(\mathbf{u}) \tag{19}
\end{aligned}
$$

based on the results in Prop. B.1.

$\square$

**Corollary B.2.1.** The partial derivative of the path density $\mathbf{p}(\mathbf{u})$ with respect to action preference $A_{ij}$ is

$$\partial_{A_{ij}} \log \mathbf{p}(\mathbf{u}) \;=\; \mathbf{p}(\mathbf{u}) \left[ n_{ij}(\mathbf{u}) - e^{A_{ij} - A_{ii_\omega}} n_{ii_\omega}(\mathbf{u}) \right] \;. \tag{20}$$

*Proof.* Using the log-derivative trick $\partial_{A_{ij}} \mathbf{p}(\mathbf{u}) = \mathbf{p}(\mathbf{u}) \partial_{A_{ij}} [\log \mathbf{p}(\mathbf{u})]$. $\qquad \square$

**Proposition B.3.** The path-expectation of the partial derivatives of $\log \mathbf{p}(\mathbf{u})$ and $\log \boldsymbol{\pi}(\mathbf{u})$ with respect to an action preference is zero:

$$\sum_{\mathbf{u} \in \mathcal{U}} \mathbf{p}(\mathbf{u}) \left[ \partial_{A_{ij}} \log \mathbf{p}(\mathbf{u}) \right] = 0 \;. \tag{21}$$

*Proof.* Proving for $\log \mathbf{p}(\mathbf{u})$

$$\begin{aligned}
\sum_{\mathbf{u} \in \mathcal{U}} \mathbf{p}(\mathbf{u}) \left[ \partial_{A_{ij}} \log \mathbf{p}(\mathbf{u}) \right] &= \sum_{\mathbf{u} \in \mathcal{U}} \mathbf{p}(\mathbf{u}) \left[ n_{ij}(\mathbf{u}) - \frac{e^{A_{ij}}}{e^{A_{ii_\omega}}} n_{ii_\omega}(\mathbf{u}) \right] \\
&= \langle n_{ij}(\mathbf{u}) \rangle_{\mathbf{p}} - \frac{e^{A_{ij}}}{e^{A_{ii_\omega}}} \langle n_{ii_\omega}(\mathbf{u}) \rangle_{\mathbf{p}} \\
&= \mathcal{C}_{ij} - e^{A_{ij} - A_{ii_\omega}} \mathcal{C}_{ii_\omega} \;.
\end{aligned} \tag{22}$$

The two-point state counter correlations $\mathcal{C}_{ij}$ can be expressed in terms of the successor representation $D$ and policy $\pi_{ij} = e^{A_{ij}}$ as $\mathcal{C}_{ij} = D_{0i} e^{A_{ij}}$. Therefore, we continue

$$\begin{aligned}
\sum_{\mathbf{u} \in \mathcal{U}} \mathbf{p}(\mathbf{u}) \left[ \partial_{A_{ij}} \log \mathbf{p}(\mathbf{u}) \right] &= -\tau D_{0i} e^{A_{ij}} + \tau e^{A_{ij} - A_{ii_\omega}} D_{0i} e^{A_{ii_\omega}} \\
&= -\tau D_{0i} e^{A_{ij}} + \tau D_{0i} e^{A_{ij}} \\
&= 0 \;.
\end{aligned} \tag{23}$$

$\qquad \square$

**Corollary B.3.1.** The partial derivative of the regularized path reward term $\mathbf{J}(\mathbf{u})$ with respect to $A_{ij}$ is

$$\partial_{A_{ij}} \mathbf{J}(\mathbf{u}) \;=\; -\tau n_{ij}(\mathbf{u}) + \tau n_{ii_\omega}(\mathbf{u}) e^{A_{ij} - A_{ii_\omega}} \;. \tag{24}$$

and its path-expectation is zero for all action preferences.

*Proof.* Since $\partial_{A_{ij}} \mathbf{J}(\mathbf{u}) = -\partial_{A_{ij}} [\log \boldsymbol{\pi}(\mathbf{u})]$. $\qquad \square$

## B.2 FISHER INFORMATION

State transition occupations are not independent. Modifying one transition occupation probability under the policy $\pi$ may change the occupation probability of another transition. This is in contrast to the expected reward objective in path space where policy modifications are independent along each path dimension (apart from an overall normalization factor). In order to identify a policy gradient in transition space with independent gradient components, we will transform the gradient derived in Section 3 into the natural path gradient pulled back to transition space. In order to make this gradient ascent natural in the space of transitions, we pre-multiply the gradient by the inverse Fisher information $\mathcal{I}^{-1}$ (Kakade, 2001) which relates the policy densities in path space $\boldsymbol{\pi}$ and transition

space $\pi$. The Fisher information matrix $\mathcal{I}$ has components

$$
\begin{aligned}
\mathcal{I}_{ij,kl} \quad &:= \quad \left\langle \left[ \partial_{A_{ij}} \log \mathbf{p}(\mathbf{u}) \right] \left[ \partial_{A_{kl}} \log \mathbf{p}(\mathbf{u}) \right] \right\rangle_{\mathbf{p}} \\
&= \quad \left\langle \left[ n_{ij}(\mathbf{u}) - e^{A_{ij} - A_{ii_\omega}} n_{ii_\omega}(\mathbf{u}) \right] \left[ n_{kl}(\mathbf{u}) - e^{A_{kl} - A_{kk_\omega}} n_{kk_\omega}(\mathbf{u}) \right] \right\rangle_{\mathbf{p}} \\
&= \quad \left\langle n_{ij}(\mathbf{u}) n_{kl}(\mathbf{u}) \right\rangle_{\boldsymbol{\pi}} - e^{A_{kl} - A_{kk_\omega}} \left\langle n_{ij}(\mathbf{u}) n_{kk_\omega}(\mathbf{u}) \right\rangle_{\mathbf{p}} + \\
&\quad - e^{A_{ij} - A_{ii_\omega}} \left\langle n_{kl}(\mathbf{u}) n_{ii_\omega}(\mathbf{u}) \right\rangle_{\mathbf{p}} + e^{A_{ij} - A_{ii_\omega}} e^{A_{kl} - A_{kk_\omega}} \left\langle n_{ii_\omega}(\mathbf{u}) n_{kk_\omega}(\mathbf{u}) \right\rangle_{\mathbf{p}} \\
&= \quad \mathcal{C}_{ij,kl} - e^{A_{kl} - A_{kk_\omega}} \mathcal{C}_{ij,kk_\omega} - e^{A_{ij} - A_{ii_\omega}} \mathcal{C}_{kl,ii_\omega} + e^{A_{ij} + A_{kl} - A_{ii_\omega} - A_{kk_\omega}} \mathcal{C}_{ii_\omega,kk_\omega} \quad .
\end{aligned}
$$
(25)

where we have used Prop. B.2. The Fisher information $\mathcal{I}$ depends on the counter correlation functions $\mathcal{C}$. The counter correlation functions can be derived using Markov chain theory (Kemeny & Snell, 1983).

## B.3 INITIALIZATION

The prior policy $\pi^0$ can be set to any stochastic policy with corresponding initial action preferences

$$
A_{ij}^0 \quad = \quad \tau \log \pi_{ij}^0 \quad .
$$
(26)

Assuming that $\pi^0$ is initialized at the random policy, we have

$$
\pi_{ij}^0 \quad = \quad \frac{1}{|\mathcal{A}_i|}
$$
(27)

$$
A_{ij}^0 \quad = \quad -\tau \log |\mathcal{A}_i|
$$
(28)

for all states $s_i \in \mathcal{S}$ and actions $a_j \in \mathcal{A}_i$.

