# OpenReview forum: "Policy path programming"
_ICLR.cc/2020/Conference — Reject_

### Official Review · AnonReviewer1 · 2019-10-18
**Official Blind Review #1**

**Rating:** 3

**Review:**

This paper presents a reinforcement learning method that exploits the full-depth backup. The policy update based on the full-depth backup is derived for entropy-regularized MDP. The state-action correlation function is introduced and the Fisher information matrix is computed with it. The proposed method is evaluated on tasks with discrete states and actions.

I understand the concept of using the full path for updating the policy, but I do not see significant novelty of the proposed method from the current manuscript. The proposed method looks equivalent to the natural policy gradient with full-depth backup for entropy-regularized MDP, which is a special case of existing methods.

My concern is that the scalability of the proposed method. The use of the full-depth backup should suffer from the large variance, and I think the proposed method will not work on tasks with the high-dimensional state space.
The evaluation is limited to tasks of which the state space is small, and the proposed method is not compared with existing methods.

Due to the unclear novelty and limited empirical results, I give weak reject to the paper in the current form.

I request authors to answer the following questions to improve the clarity.

- Is the proposed method equivalent to use the natural policy gradient with the full-depth backup for a softmax energy-based policy? If they are different, what it the crucial difference?

- I think that the variance of the estimation of the gradient is large when using the full-depth back up. I'm curious about the performance of the proposed method in high-dimensional tasks. However, the evaluation is very limited to simple tasks in which the state space is relatively small compared with tasks commonly used in deep RL papers.
Does the proposed method scale to more complex tasks, such as Atari games?

- When using the n-step TD learning, increasing n does not always improve the performance, and n should be set to an intermediate value
What is the motivation of using the full paths for updating the policy? Does the proposed method outperform existing methods? Especially, the comparison with natural policy gradient methods is necessary to show the benefit of the proposed algorithm.

Minor comments:

- In page 2, "A state-space \mathcal{X} is composed of states x \in \mathcal{X}" <- Authors may want to replace x with s in this sentence.
 - In page 6, "0\lambda < 1" I think that "<" is missing between "0" and "\lambda"
- In page 6, some variables are explained after Equation (10). However, it took me a while to find "lambda" in Equation (10), since Equation (10) has 9 lines and many terms. I think the description in page 6 can be improved. For example, I recommend to use "\exp" instead of "e^" for readability.

**Experience Assessment:**

I have published one or two papers in this area.

**Review Assessment: Checking Correctness Of Derivations And Theory:**

I assessed the sensibility of the derivations and theory.

**Review Assessment: Checking Correctness Of Experiments:**

I assessed the sensibility of the experiments.

**Review Assessment: Thoroughness In Paper Reading:**

I read the paper at least twice and used my best judgement in assessing the paper.

---

### Official Review · AnonReviewer4 · 2019-10-23
**Official Blind Review #4**

**Rating:** 3

**Review:**

This work proposes a policy iteration algorithm that implements full-depth, full-width backups in contrast to one-step, full-width methods. The authors go over existing algorithms and talks a bit how their proposal conceptually differs in how it performs said backups. They provide a bit of intuition to help explain their algorithm's derivation. Finally, they provide a few experiments showing that their algorithm works.

My personal issues are with these experiments. First, I would like to see better comparisons between this method and existing policy iteration methods. I don't have a good sense in which one would choose to use this algorithm over any baseline methods. Is it faster in any sense? Does it produce better policies during certain games? For the experiments themselves, I don't see much clarification of what the various graphs even show. More effort should have been spent analyzing these.

I come away from this work not fully appreciating the impact it is trying to sell me on. I also think the discussion section should have been more fleshed out.

**Experience Assessment:**

I do not know much about this area.

**Review Assessment: Checking Correctness Of Derivations And Theory:**

I assessed the sensibility of the derivations and theory.

**Review Assessment: Checking Correctness Of Experiments:**

I assessed the sensibility of the experiments.

**Review Assessment: Thoroughness In Paper Reading:**

I read the paper at least twice and used my best judgement in assessing the paper.

---

### Official Review · AnonReviewer3 · 2019-10-27
**Official Blind Review #3**

**Rating:** 3

**Review:**

The paper considers the problem of entropy-regularized discounted Markov decision processes with discrete state space. Instead of working on the parameter space of policy (\pi_ij), the paper has proposed to reparametrize with natural parameters (A_ij). The reparameterization trick helps to learn the natural parameters using the natural gradient method.
The writing is easy to follow. However, it is not clear what is the benefit of learning policy using path representation compared with other methods in the literature. The paper does not clearly state the motivation of the proposed method.
The experimental section presents the convergence of the proposed methods in 3 small problems including decision trees of four levels, the tower of Hanoi problem, and four-room grid worlds. The experiment setting is very simple with a small number of states in the policy. It is not clear how the proposed method is able to scale up the size of state space. Besides, there is no baseline method in the literature to be presented to compare with the proposed method.


**Experience Assessment:**

I do not know much about this area.

**Review Assessment: Checking Correctness Of Derivations And Theory:**

I did not assess the derivations or theory.

**Review Assessment: Checking Correctness Of Experiments:**

I did not assess the experiments.

**Review Assessment: Thoroughness In Paper Reading:**

I read the paper at least twice and used my best judgement in assessing the paper.

---

### Official Review · AnonReviewer2 · 2019-11-03
**Official Blind Review #2**

**Rating:** 1

**Review:**

The paper considers the problem of finding the optimal policy in the Markovian decision Processes, where a KL policy regularizer is added to the objective function. Instead of the closed form solution which leads  to the KL-regularized Bellman equation the paper proposes to use an incremental gradient ascent algorithm. The paper recommends an iterative policy gradient scheme to  optimize this  objective function. There exists a substantial literature on the subject of KL-regularized RL as well as using the  policy gradient algorithms  to optimize this objective function using policy gradient schemes (See all variants of KL(entropy)-constraint actor-critic or reinforce algorithms, e.g. A2C, IMPALA,...). Unfortunately the paper doesn’t provide any comparison with those methods. In the absence of those comparisons the significance of this work to the literature of RL is not clear, as it is not solving an open problem which hasn't addressed before, neither it  provides theoretical/empirical evidence that it has advanced the start-of-the-art in terms of providing a more efficient solution.

 The paper considers a setting which is quite well-studied as there exist efficient solvers for optimizing the KL regularized RL (including policy-gradient variants). Why the proposed approach is better than those already existed in the literature? What is the outstanding problem in the literature of KL-regularized RL that this work tries to address? I couldn’t find a satisfying argument with respect to these questions in the current submission.    In the absence of any theoretical or empirical result to justify the merits of the proposed algorithm the contribution of this paper to the literature is not clear. Also it is not clear how this approach can scale up to anything beyond the finite state-action  problems as it relies on knowing quantities like state-action transitions and the inverse of state transition matrix which in practice is quite difficult to estimate. I recommend the authors to rethink their approach from the point of view of whether  It provides solution to some open problems in RL/control or it advances the-state-of-the-art. If this is the case, the paper needs to provide theoretical/empirical evidence to back up its claim. Unfortunately the current submission does not satisfy these requirements.


**Experience Assessment:**

I have published in this field for several years.

**Review Assessment: Checking Correctness Of Derivations And Theory:**

I assessed the sensibility of the derivations and theory.

**Review Assessment: Checking Correctness Of Experiments:**

I assessed the sensibility of the experiments.

**Review Assessment: Thoroughness In Paper Reading:**

I read the paper at least twice and used my best judgement in assessing the paper.

---

### Author Response · Authors · 2019-11-14
**Author response**

Thanks to all reviewers for your feedback. The manuscript has been edited to address some of the issues raised and to improve its clarity and precision. The overall impression is that comparative demonstrations of this theory embedded in a scalable RL algorithm is required which is not possible at this stage. To address Reviewer #1's questions directly:

- No. In fact, the natural path gradient could be combined with the natural policy gradient through the reparameterization rule for Fisher informations. Concretely, this would result in policy parameter updates sensitive to state-action correlations under the policy-induced path distribution (which is not the case with the natural policy gradient).
- Untested as yet.
- To do an exact full-width, full-depth backup via roll-outs is impossible since this would require an infinite number of infinitely deep samples in general. Our model accomplishes this using a known environmental model (and exhibits hierarchical processing of the environment and policy dynamics). With respect to an implementation in a scalable RL agent, though untested, our method suggests an alternative approach by which a full-depth (or n-step) backup may be approximated based on estimating the components of the path gradient calculation during exploration. This approach has the additional benefit that estimated path gradient components transfer across reward functions.

---

### Decision · Program_Chairs · 2019-12-19

**Decision:**

Reject

**Comment:**

The reviewers were not convinced about the significance of this work. There is no empirical or theoretical result justifying why this method has advantages over the existing methods. The reviewers also raised concerns related to the scalability of the proposal. Since none of the reviewers were enthusiastic about the paper, including the expert ones, I cannot recommend acceptance of this work.